# CT-Guided vs. Navigational Bronchoscopic Biopsies for Solitary Pulmonary Nodules: A Single-Institution Retrospective Comparison

**DOI:** 10.3390/cancers15215258

**Published:** 2023-11-02

**Authors:** Fawad Aleem Chaudry, Maureen Thivierge-Southidara, Juan Carlos Molina, Samid M. Farooqui, Syed Talal Hussain, Moishe Libermen

**Affiliations:** 1Department of Thoracic Surgery, Centre Hospitalier de l’Université de Montréal (CRCHUM), Montreal, QC H2X 0C1, Canada; fawad-chaudry@ouhsc.edu (F.A.C.);; 2Department of Pulmonary and Critical Care Medicine, University of Oklahoma Health Sciences Center, Oklahoma City, OK 73117, USA; samidmf@outlook.com; 3Faculty of Education Sciences, Faculty of Medicine, Université de Montréal, Montreal, QC H2X 0C1, Canada; 4Faculté de Médecine, Université Laval, Québec City, QC G1V 4G2, Canada; 5Department of Surgery, Université de Montréal, Montreal, QC H2X 0C1, Canada; 6Research Centre, Centre Hospitalier de l’Université de Montréal (CRCHUM), Montreal, QC H2X 0A9, Canada

**Keywords:** electromagnetic navigational bronchoscopy, lung cancer, pulmonary nodule, transthoracic needle aspiration

## Abstract

**Simple Summary:**

Since the advent of lung cancer screening, the incidence of finding lung nodules has increased. Biopsy of these nodules can be performed by various techniques. In this study we have compared diagnostic accuracy and safety profile of two commonly used techniques; Electromagnetic navigational bronchoscopy (ENB) and CT-guided transthoracic needle aspiration (TTNA). In this retrospective study, we show that CT-guided TTNA has a higher diagnostic accuracy but also higher rate of complications as compared to ENB.

**Abstract:**

Objective: Lung cancer is the second most common cause of death by cancer. Multiple modalities can be used to obtain a tissue sample from a pulmonary nodule. We aimed to compare the yield and adverse events related to transthoracic needle aspiration (TTNA) and Electromagnetic Navigation Biopsy (ENB) at our institution. Methods: This was a single-center retrospective study in which all patients referred for evaluation of a pulmonary lesion over 5 years (1 January 2013 to 31 December 2018) were identified. Our primary outcome was to compare the accuracy of TTNA to that of ENB in establishing the diagnosis of pulmonary lesions. Secondary outcomes included the evaluation of the adverse events and the sensitivity, specificity, positive, and negative predictive value of each modality. Results: A total of 1006 patients were analyzed. The mean age of patients in the TTNA and the ENB group was 67.2 ± 11.2 years and 68.3 ± 9.2 years respectively. Local anesthesia was predominantly used for TTNA and moderate sedation was more commonly used in the ENB group. We found ENB to have an accuracy of 57.1%, with a sensitivity of 40.0%, a specificity of 100.0%, a positive predictive value of 100.0%, and a negative predictive value of 40.0%. As for the TTNA, the accuracy was 75.9%, with a sensitivity of 77.5%, a specificity of 61.5%, a positive predictive value of 95.0%, and a negative predictive value of 22.5%. The rate of clinically significant complications was higher in the TTNA group (8.2%) as compared to the ENB group (4.7%) with a *p*-value < 0.001. Conclusion: TTNA was superior to ENB-guided biopsy for the diagnostic evaluation of lung nodules. However, the complication rate was much higher in the TTNA group as compared to the ENB group.

## 1. Introduction

Lung cancer is the second most common cancer and accounts for the highest number of cancer-related mortality [1]. In 2011, the National Lung Screening Trial (NSLT) showed a 20% decrease in lung cancer-specific mortality from screening at-risk populations using Computed Tomography (CT) scanning [2]. Since then, the screening of at-risk individuals has become the standard of care, which increased incidental findings of lung nodules requiring further evaluation and workup. 

Once a pulmonary nodule is identified, further steps need to be taken based on the patient’s risk factors and the pre-test probability of lung cancer [3]. These steps include (i) radiological follow-up with serial CT scans, (ii) sampling of the lesion for pathological diagnosis, and (iii) surgical resection. 

When a decision is made to biopsy a lung nodule, many modalities can be used. CT-guided transthoracic needle aspiration (TTNA) is a commonly used technique due to its high diagnostic yield rate, but this modality is also associated with an increased risk of complications, such as pneumothorax [3,4]. Flexible bronchoscopy can also be used, but the yield is quite low, varying from 14% to 30%, based on multiple lesion characteristics, and the operator’s experience [5]. Guided bronchoscopy with the application of different techniques has varied diagnostic yield rates. These techniques include electromagnetic navigation biopsy, virtual bronchoscopy (VB), radial probe endobronchial ultrasound (RP-EBUS), ultrathin scope combined with RP-EBUS, and multi-modality (a combination of several of the aforementioned techniques) [6]. According to our knowledge, there has been no head-to-head analysis performed between CT-guided TTNA and ENB for the evaluation of peripheral pulmonary nodules. This study aspires to provide a better understanding of the utility of these techniques. 

## 2. Materials and Methods

This is a retrospective study of data from a single tertiary care institution. This study was approved by ‘CHUM REB’, the Institutional Review Board at our institution (IRB number #17.064). Patients referred for the biopsy of a pulmonary nodule were identified using a prospectively maintained database. The inclusion criteria included the patients who underwent TTNA or ENB from 1 January 2013 to 31 December 2018. Patients under the age of 18 and those with more than 20% of missing data were excluded. 

Chart review and data collection were carried out by two researchers. The information recorded from the charts included demographic imaging characteristics of the nodule (i.e., size, lobe distribution, central or peripheral location from the pleura, and Fludeoxyglucose uptake). Data on the sampling technique were collected, such as the modality used, type of sedation, use of fluoroscopy, sampling technique, rapid on-site evaluation (ROSE), and the number of biopsies. Complications were compiled in addition to their grade according to the extended Clavien-Dindo classification [7] and the interventions required. The quality of the biopsy specimen (i.e., positive or negative for cancer, inadequate specimen, positive for a non-neoplastic process) and histopathological results (i.e., granuloma, cancer, metastasis, lymphoma, other) were also collected. In order to calculate the diagnostic testing accuracy for cancer (sensitivity, specificity, yield, predictive values, and accuracy), a confirmation of pathology was obtained using subsequent histopathological outcomes through either surgery, TTNA, or bronchoscopy. If no confirmation histopathology was available, subsequent imaging was used to determine if the nodule was a neoplastic process according to the progression, based on the expert review by the radiologists. Subsequent histopathological imaging was collected within the following year of the initial procedure. 

## 3. Statistical Analysis

Statistical analyses were performed using the IBM Statistical Package for the Social Sciences (SPSS, version 25.0) [8]. When relevant, Cohen’s d effect size was measured using Lenhard and Lenhard’s calculator [9]. Mean values are presented as means  ±  standard deviation (SD), and categorical variables are presented as numbers and percentages. The differences between TTNA and ENB were assessed using chi-squared tests and Fisher’s exact tests when the expected count was less than 20% for categorical variables, and Student’s t tests were used for continuous variables. Accuracy was defined as the proportion of patients in which ENB or TTNA yielded a true diagnosis. ENB and TTNA performance for cancer diagnosis was evaluated using sensitivity, specificity, and positive and negative predictive values. As these diagnostic testing tools require a clear result (i.e., positive or negative) and a “gold standard” confirmation test to calculate, patients whose initial biopsy yielded an inadequate specimen or who did not have a confirmation test were excluded from the calculations. The level of significance for the analysis was set at *p* < 0.05. 

## 4. Results

### 4.1. Study Sample

A total of 1012 patients underwent a lung nodule biopsy between 2013 and 2018, using either TTNA or ENB. Six patients were removed as more than 20% of the data was missing from their charts; only one of them was from the TTNA group. Therefore, a total of 1006 patients were included in the analysis. Of those, 900 patients (89.5%) underwent TTNA and 106 (10.5%) underwent ENB (Figure 1). There were no differences in sex or age between the groups (Table 1). 

The biopsied nodules were bigger in the TTNA group. However, the effect size of this difference was small (Cohen’s d = 0.391) (Table 1). There were also more nodules with a diameter larger than 2 cm in the TTNA group compared to the ENB group. The mean SUV max and the number of nodules with a metabolism superior to 2.5 were equally distributed between the groups. The location from the pleura (central vs. peripheral) of the biopsied nodule was missing for 75.1% of patients in the ENB group, thus this data is not included in the results. However, the distribution of the nodules in different lobes was available, which is displayed in Table 1. There were more nodules that were biopsied in the left lower lobe using TTNA than ENB (Table 1).

### 4.2. Procedure

The anesthesia used for the procedures was different between the groups (Table 2). The majority of the TTNA biopsies were under local anesthesia, and none required general anesthesia. On the contrary, most ENB required moderate sedation and a minority required general anesthesia. Electromagnetic navigational bronchoscopy was performed using the SPiN Thoracic Navigational System by Veran Medical Technologies (St. Louis, MO, USA) (Figure 2). ENB biopsies were obtained using an aspiration needle, cytology brush, and biopsy forceps. Cryo-biopsies were not performed. For the ENB sample, ROSE was not used, but a majority of patients required fluoroscopy. Due to missing information, data regarding the use of radial endobronchial ultrasound (r-EBUS) were not collected. The mean number of biopsies required to obtain a sample was (5.35 ± 2.62) (Table 2). Similarly, neither ROSE or a frozen section were available to the TTNA group.

Even though the number of complications was higher in the TTNA group, the majority of them did not require an intervention. The type of complication that occurred was also different between the groups (Table 2). Complications associated with TTNA were air leaks, such as pneumothorax or pneumomediastinum, and bleeding, such as peri-alveolar bleeding, hemoptysis, and hemothorax. Most leaks presented as pneumothorax (n = 411; 86.2%); the remaining presented as pneumomediastinum (n = 3; 0.6%) and one patient presented with a combination of pneumothorax and pneumomediastinum (n = 1; 0.2%). Most air leaks only required observation (82.2%, n = 341), although some required more invasive procedures (17.8%, n = 74) such as chest tubes, identified as a grade 3a complication according to the Clavien-Dindo classification. Also, most bleeding complications did not require advanced treatment (91.1%, n = 102). The remaining required an intervention such as a chest tube (8.1%, n = 9). Other complications included a cough, which was resolved with oxygen, and bradycardia, which did not require an intervention.

As for patients undergoing the ENB procedure, only one patient suffered from an air leak that presented as bilateral tension pneumothorax, which required bilateral chest tube placement. Unfortunately, the patient died as a result of this complication. One patient suffered from an opioid overdose, which was resolved with naloxone. Because of the seriousness of this complication, it was identified as a grade 4a. No bleeding induced by ENB required an intervention. Other complications induced by ENB were coughing (n = 14), which did not require an intervention, and desaturation, which required oxygen for the majority (n = 3) of patients (grade 2) (Table 2).

### 4.3. Histopathology

A higher proportion of positive biopsies was found in the TTNA group. Subsequent confirmation information was only identified in 321 (35.7%) patients (Table 3). As previously stated, patients (n = 51) whose biopsy resulted in an inadequate specimen were removed from the analysis. This allowed us to calculate a sensitivity of 77.5%, a specificity of 61.5%, a positive predictive value of 95.0%, a negative predictive value of 22.5%, and an accuracy of 75.9%. The modalities used as “gold standards” were surgery (n = 198, 73.3%), CT-guided TTNA (n = 39, 14.4%), and bronchoscopy (n = 33, 12.2%). If we consider that all the inadequate specimens were actually true negatives, then this will result in a calculated sensitivity of 77.5%, a specificity of 87%, a positive predictive value of 95%, a negative predictive value of 55%, and an accuracy of 80%. If all of the inadequate specimens are considered to be false negatives, then this will yield a sensitivity of 64%, a specificity of 62%, a positive predictive value of 95%, a negative predictive value of 13%, and an accuracy of 64%. 

For the ENB group, subsequent confirmation information was only identified in 71 (67.0%) patients. One patient’s biopsy resulted in an inadequate specimen, which was later confirmed as a negative result through follow-up imaging and was removed from the following analysis. This resulted in a sensitivity of 40.0%, a specificity of 100.0%, a positive predictive value of 100.0%, a negative predictive value of 40.0%, and an accuracy of 57.1%. The modalities used as a “gold standard” were surgery (n = 7, 10.0%), CT-guided TTNA (n = 4, 5.7%), bronchoscopy (n = 17, 24.3%), and follow-up imaging (n = 42, 60.0%) (Table 3). 

Test characteristic differences between CT-guided TTNA and ENB remained when the data were separated between larger and smaller nodules. Hence, specificity and positive and negative predictive values were higher in the ENB group. On the contrary, sensitivity and accuracy were higher in the CT-guided TTNA group. All those values were higher when calculated for larger nodules (Table 3).

The most common histopathology that was found in the biopsies was primary lung cancer in both groups. Other results in the CT-guided TTNA group included solid and hematological metastasis, granulomas, inflammatory lesions, benign tumors, pneumoconiosis, and infectious etiologies (Table 4). 

In the ENB group, other results included biopsies suspicious for malignancy and infectious etiologies. Primary lung cancer in this group presented as either adenocarcinoma (n = 16, 34.0%), squamous cell carcinoma (n = 17, 36.2%), undifferentiated non-small cell lung cancer (n = 4, 8.5%), or neuroendocrine carcinoma (n = 1, 2.1%).

## 5. Discussion

With increased screening for lung cancer in at-risk individuals, there is a need to sample suspicious lung nodules in a safe manner. The TTNA of a lung nodule has been reported to have high diagnostic yield rates, with Lee et al. reporting a sensitivity of 92.5% [10], but it is associated with a high rate of complications, especially pneumothorax [4,11,12]. ENB is a technique that allows for the identification of a pathway to the pulmonary lesion by constructing a three-dimensional image of the lung anatomy using CT scan images and sensor location [13]. A yield of 64.9% was reported by Gex et al. in their meta-analysis [14]. The 12-month results from the NAVIGATE Trial reported a 12-month diagnostic yield of 73% with pneumothorax, requiring chest tube placement or admission, only occurring in 2.9% of patients [15]. Other studies have shown varying diagnostic yield rates for ENB between 59% and 77% [16,17]. We tried to compare these two outcomes of the two modalities at our facility. 

There was no difference in the baseline characteristics of the populations in the two groups in our study. The size of the nodule in the CT TTNA group was larger than in the ENB group, but the effect size of the difference was small. There was a statistically significant difference in the left lower lobe nodules biopsied via CT TTNA than with ENB. This is important to note, as a previous study showed that the diagnostic yield of ENB significantly drops if the lesion is located at the lower lobes [18]. In our study, approximately 20% of biopsied nodules eventually underwent surgical resection, which may be an underreported number, as a lot of patients were referred from peripheral centers and we may not have had an accurate assessment of surgical resection rates based on retrospective review. 

Requirements for the type of anesthesia were significantly different in our two groups. Most of our CT TTNA group patients had the procedure performed under local anesthesia, which was different from the type of sedation used by McSweeney et al. [19] when they analyzed percutaneous lung biopsies at their center. On the contrary, the majority of the ENB procedures were performed under sedation. This was similar to the experience of Sato et al. [20]. Interestingly Bertoletti et al. used an inhalational mixture of a 50/50 combination of nitrous oxide/oxygen for performing ENB and found that to be efficacious and well tolerated [21].

Our study showed that clinically significant complications, classified as Clavien-Dindo Class II and above, occurred more frequently in the CT TTNA group (8.2%) compared to the EMB group (4.7%). Our findings re-establish the high occurrence of pneumothoraces after a CT-guided biopsy; however, it must be noted that most of these were managed with observation and were not clinically significant. If we included these small pneumothoraces, which did not require any intervention in our analysis, we yielded similar complication rates to previous studies [11,22]. Also, the slightly higher rate of pneumothorax in our study may reflect the possibility that, in most institutes, pneumothoraces that occur during the procedure are manually aspirated before the introducer needle is removed [23], which may not be the practice at our center. It is important to note there were no deaths in the TTNA group. 

The distance of the lesion from the periphery has been shown to be an important risk factor for the development of pneumothorax during CT-guided biopsy [24]. Unfortunately, we could not include this parameter in our analysis, given the large amount of missing data. Dibardino et al. [25] reported a 20% decrease in the incidence of pneumothorax if ultrasound (US)-guided needle aspiration is performed rather than CT-guided; however, this can be explained by the fact that US-guided biopsies are only performed on nodules that have direct contact with the pleura.

In our analysis, we were able to measure an accuracy of 57.1% for the ENB group, with a sensitivity of 40.0% and a specificity of 100.0%. This was lower than the accuracy and sensitivity recently reported by Cheng et al. [26] (71.7% and 67.8%, respectively). The NAVIGATE trial [15] also reported a sensitivity of 69% for ENB after one year. This reflects the variability associated with the expertise of the person performing the procedure. In a study performed by Makris et al. [27], the authors concluded that approximately 15 procedures are needed before obtaining procedural proficiency. The accuracy of the CT TTNA group was in accordance with the Society of International Radiology Guidelines, which suggests a threshold yield of 75% [28].

This study demonstrates that we are not as accurate with guided biopsy as we presumed. In our single-center review of CT-guided and bronchoscopic-guided biopsies, we yielded much lower sensitivity and accuracy as opposed to the published literature. There were multiple factors influencing this result, including operator expertise, the size of the nodule, the location and distance from the pleura (not accounted for in the current study), the use of conscious sedation instead of general anesthesia, and the non-availability of bronchus sign. A previous study by Seijo et al. showed that the diagnostic yield of electromagnetic navigation bronchoscopy is highly dependent on the presence of a Bronchus sign on CT imaging [29].

Regarding the results of our study, CT-guided TTNA was superior to ENB-guided biopsy for the diagnostic evaluation of lung nodules. However, the complication rate was much higher in the CT-guided group compared to the ENB group, which is consistent with the rates reported in the literature. 

Currently, there are no fixed guidelines regarding the diagnostic evaluation of solitary pulmonary nodules, and the availability of all diagnostic modalities (ENB, radial EBUS, navigation) varies in different institutions, which accounts for the different yield rates in different institutions. Though the accuracy and sensitivity of ENB were lower than that of CT TTNA, combining Endobronchial Ultrasound (EBUS) with ENB has yielded much better yield and accuracy rates than ENB alone [30]. Currently, the All In One Trial [31] is underway and will provide insightful information on the outcomes of these modalities. 

The strength of our study is that we presented the data of 1006 consecutive patients, which is a large number studied for the comparison between these modalities, and will help in contributing to the literature on this topic. This is the first, direct head-to-head comparison of CT-guided and guided bronchoscopic biopsies in a single institution, which will add to the growing literature regarding the diagnostic evaluation of pulmonary nodules and will pave the way for a future randomized trial comparing the two techniques. 

Our study also has some potential limitations. The absence of a true gold standard and the absence of a subsequent confirmation biopsy in a significant proportion of patients weakened our analysis, which impairs the validity of this study. As this retrospective study was conducted in a reference center, many patients underwent the procedure in our center without follow-up, as follow-ups were conducted at their treating center. This explains the high portion of the loss to follow-up. The use of a retrospective design introduced a verification bias, as negative values were more likely to have a confirmatory test if the clinical suspicion remained high. In addition, as the results of the biopsies were known to the clinicians, the decision to perform a confirmatory test or to follow the patient was not a blinded decision, thus introducing a test review bias. The direction in which this affected the accuracy of our results remains unknown due to the assortment of biases. This was a single-center, retrospective study, which affected the strength of the study. We did not collect information regarding the additional use of a radial EBUS probe in the diagnostic evaluation of patients in the ENB group, which may have led to different results had it been included. The distance of the nodule from the pleura was not calculated. Also, the data regarding the proportion of ground glass opacities among the nodules was not collected, which could affect the results. Most of the ENB procedures were performed under moderate sedation rather than general anesthesia. Data regarding needle size were not collected. For both groups, the adequacy of the specimens was collected for tumor identification. We did not collect if specimens were adequate for tumor molecular studies.

## 6. Conclusions

To conclude, based on our results, we believe that guided biopsy techniques are not currently as accurate as they were perceived to be, based on the previous literature. We propose similar head-to-head studies, in a prospective manner, to gain a better understanding of the role of guided biopsies in the evaluation of peripheral lung nodules. 

## Figures and Tables

**Figure 1 cancers-15-05258-f001:**
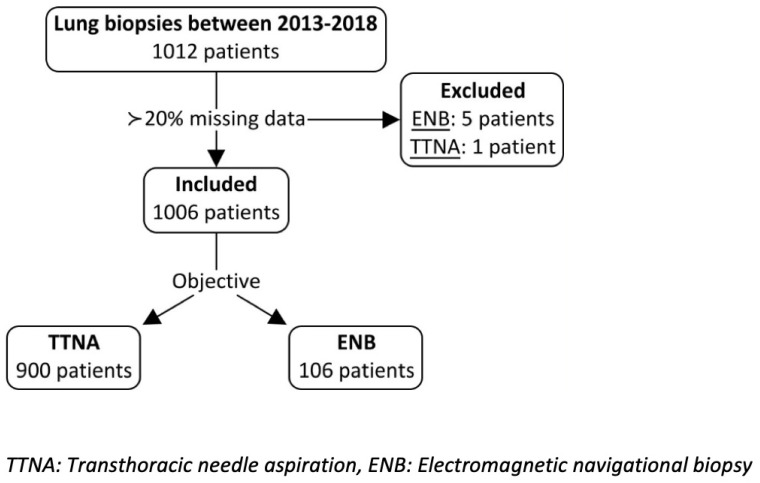
Study enrollment: Six patients had more than 20% missing data and were excluded from the study enrollment.

**Figure 2 cancers-15-05258-f002:**
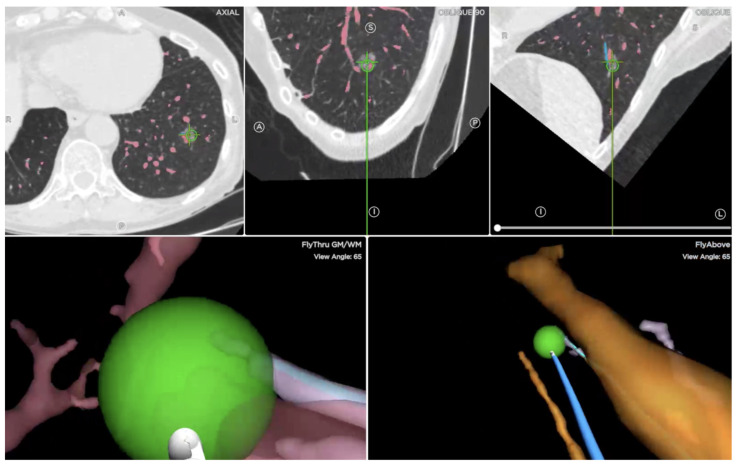
Representative screen capture during Electromagnetic Navigational Bronchoscopy showing axial and oblique views to locate and biopsy the target lesion.

**Table 1 cancers-15-05258-t001:** Patient and lesion characteristics.

		TTNA	ENB	Probability (*p*)
N (%)		900 (89.5)	106 (10.5)	------------------
Female sex (%)		504 (56.0)	56 (52.8)	0534
Mean age (years) ± SD		67.2 ± 11.2	68.3 ± 9.2	0.250
Diameter (mm)	Mean ± SD	22.83 ± 15.34	17.03 ± 9.20	<0.001
Nodule diameter ≥ 2 cm n (%)		499 (55.9)	39 (38.2)	0.001
SUV max	Mean ± SD	6.97 ± 9.64	5.90 ± 10.33	0.314
Nodule SUV max ≥ 2.5 n (%)		629 (70.0)	60 (63.8)	0.213
Nodule location n (%)	LLL	187 (20.8)	10 (9.4)	0.005
	Lingula	16 (1.8)	5 (4.7)	0.061
	LUL	209 (23.2)	32 (30.2)	0.112
	RLL	172 (19.1)	20 (18.9)	0.952
	RML	44 (4.9)	9 (8.5)	0.116
	RUL	272 (30.2)	30 (28.3)	0.683

LLL: Left lower lobe, LUL: Left upper lobe, RLL: Right Lower Lobe, RML: Right Middle Lobe, RUL: Right Upper Lobe.

**Table 2 cancers-15-05258-t002:** Comparison of complications between TTNA and ENB groups.

			TTNA	ENB	Probability (*p*)
Anesthesia n (%)	Local		727 (99.2)	0 (0.0)	<0.001
	Sedation		6 (0.8)	84 (84.8)	
	General		0 (0.0)	14 (15.2)	
Use of fluoroscopy n (%)			---------	65 (61.3)	---------
Use of ROSE n (%)			---------	0 (0.0)	---------
Mean number of biopsies (± SD)			---------	5.35 (±2.62)	---------
Number of patients with complications n (%)			477 (53.0)	20 (18.9)	<0.001
Clavien-Dindo classification per patient with complications n (%)	Grade 1		403 (84.5)	15 (75.0)	0.264
	Grade 2		0 (0.0)	3 (15.0)	
	Grade 3a		74 (15.5)	0 (0.0)	
	Grade 4a		0 (0.0)	1 (5.0)	
	Grade 5		0 (0.0)	1 (5.0)	
Complication types n (%) Percentages represent % of complications from total complications	Air leak		415 (87.0)	1 (4.0)	<0.001
	Pneumothorax		411 (86)	1 (4.0)	<0.001
	Bleeding		111 (23.3)	7 (28.0)	0.227
	Other	Total	1 (0.2)	17 (68.0)	<0.001
		Cough	1 (0.2)	11 (44.0)	---------
		Bradycardia	1 (0.2)	0 (0.0)	---------
		Desaturation	0 (0.0)	6 (24.0)	---------
Intervention required per patient n (%)	Total		82 (17.2)	6 (30.0)	<0.001
	Chest tube		74 (90.2)	0 (0.0)	---------
	Aspiration		8 (9.8)	1 (16.7)	---------
	Other	Total	0 (0.0)	5 (83.3)	---------
		Oxygen	0 (0.0)	3 (50.0)	---------
		Naloxone	0 (0.0)	1 (16.7)	---------
		Advanced cardiovascular life support (ACLS)	0 (0.0)	1 (16.7)	---------

**Table 3 cancers-15-05258-t003:** Biopsy results.

		Sensitivity (%)	Specificity (%)	Positive Predictive Value (%)	Negative Predictive Value (%)	Accuracy (%)
CT-guided TTNA	<2 cm	63.0	75.0	95.4	19.6	64.3
	≥2 cm	87.4	46.2	94.7	25.0	84.0
	Overall	77.5	61.5	95.0	22.5	75.9
ENB	<2 cm	34.4	100.0	100.0	40.0	54.3
	≥2 cm	56.3	100.0	100.0	41.7	66.7
	Overall	40.0	100.0	100.0	40.0	57.1

**Table 4 cancers-15-05258-t004:** Biopsy results with sensitivity for patients with cancer.

		CT Guided TTNA	ENB	Probability (*p*)
Pathology	Negative	180 (20.0)	58 (54.7)	<0.001
	Positive	624 (69.3)	47 (44.3)	
	Inadequate specimen	96 (10.7)	1 (0.9)	
	Other	120 (19.2)	9 (19.1)	
Sensitivity for cancer patients		80.3	39.5	------------------

## Data Availability

The data that support the findings of this study are available from University of Montreal but restrictions apply to the availability of these data, which were used under license for the current study, and so are not publicly available. Data are however available from the authors upon reasonable request and with permission of University of Montreal.

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
