# Peer review of "CT-Guided vs. Navigational Bronchoscopic Biopsies for Solitary Pulmonary Nodules: A Single-Institution Retrospective Comparison"

_cancers, 2023, doi:10.3390/cancers15215258_

Round 1

Reviewer 1 Report

Comments and Suggestions for Authors

This is a strong manuscript on a very important topic. The authors have done an excellent job summarizing their results and appear to have been open and frank about potential weaknesses of their methods. I recommend publication of this valuable manuscript. I do have a few questions and comments for the authors. Why did they choose the endpoint of 2018 for analysis of their results? Given their stated beliefs regarding learning curves for complex methods like TTNBx and ENB, is it not expected and possible that their more recent results are substantially better. I am impressed that they had no deaths in 900 TTNBx. I would ask them to specifically state this in the text. If mortality risk of ENB is 1%, then 9 deaths might have been anticipated given equal n in both arms?

The biggest problem about the applicability of this data to modern LC screening is that in screened patients, the nodules found are considerably smaller. For example, in IELCAP, mean diameter of nodules on baseline screen are in the range of 1.5 cm and annual repeat screen nodules are less than 1 cm. What kind of results did the authors experience in such nodules. Perhaps this is an appropriate question for their next iteration of this line of investigation. 

It is of some interest that only a small proportion of those biopsied went on to surgical resection. It appears to me that this number is only about 20%? The authors might comment on this in their discussion section. 

One final issue that is probably beyond the scope of this article is the question of what features of nodule type and location that might lead to selection of TTNBx vs. ENB as the initial diagnostic modality.  Yet another potential source of analysis for future research efforts of this group of investigators. 

Finally, I would suggest that presentation of this data at an upcoming IELCAP research conference would be valuable. 

In table 3 the authors should clearly identify which data is more or less than 2 cm in diameter. 

Author Response

We wish to thank the reviewer for the insightful review and feedback. We are delighted that the reviewer finds our manuscript clinically relevant.

Reviewer :  Why did they choose the endpoint of 2018 for analysis of their results? Given their stated beliefs regarding learning curves for complex methods like TTNBx and ENB, is it not expected and possible that their more recent results are substantially better

Response: We agree that given the learning curves, there may be some differences in the results with time. Recently new diagnostic modalities like robotic navigational bronchoscopy have been introduced at our institute and some other thoracic/interventional pulmonary centers. Given that we have been moving towards exploring these modalities as well and exploring early data with newer technology. However, at most institutes in the US and across different countries, electromagnetic navigational platforms (ENB) are still being used. Our study is the first to compare TTNA vs ENB and provides guidance about diagnostic yields. 

Reviewer: I am impressed that they had no deaths in 900 TTNBx. I would ask them to specifically state this in the text. If mortality risk of ENB is 1%, then 9 deaths might have been anticipated given equal n in both arms?

Response: We agree that this is an important result and have added it in our text as recommended by reviewer.

Reviewer: The biggest problem about the applicability of this data to modern LC screening is that in screened patients, the nodules found are considerably smaller. For example, in IELCAP, mean diameter of nodules on baseline screen are in the range of 1.5 cm and annual repeat screen nodules are less than 1 cm. What kind of results did the authors experience in such nodules. Perhaps this is an appropriate question for their next iteration of this line of investigation. 

Response: With electromagnetic navigational system (ENB), it was standard at the time to sample nodules close to 1.5-2 cm. That is we have also stratified our data with that cutoff. Newer modalities like robotic bronchoscopy are showing promise with such smaller nodules (<1 cm).

Reviewer: It is of some interest that only a small proportion of those biopsied went on to surgical resection. It appears to me that this number is only about 20%? The authors might comment on this in their discussion section. 

Response: We have now commented on this in the discussion section. 

Reviewer: One final issue that is probably beyond the scope of this article is the question of what features of nodule type and location that might lead to selection of TTNBx vs. ENB as the initial diagnostic modality.  Yet another potential source of analysis for future research efforts of this group of investigators. 

Response : We agree with the reviewer that with our study and future prospective studies on this topic we may improve patient selection for ENB vs TTNA.

Reviewer: Finally, I would suggest that presentation of this data at an upcoming IELCAP research conference would be valuable. 

Response: That is an excellent suggestion and we would prepare to present this data at the conference.

Reviewer 2 Report

Comments and Suggestions for Authors

Congratulations to your great work.

However, I have several questions.

1. Based on the clinical practice, ENB can be used for peripheral or central lung nodules. But, CT-guided needle biopsy is only suggested to use for peripheral lung nodule. Therefore, please provide your data about the location of lung nodules.

2. According to review of literature and my clinical experiences, the sensitivity and specificity of CT-guided needle biopsy were low in ground-glass opacity nodules and small size. Therefore, please clarify the ground-glass opacity of lung nodules in your patients between two groups. I suggested that you should consider the impact of GGO ratio.

Author Response

We would like to thank the reviewer for their excellent feedback. Following are our comments

Reviewer : Based on the clinical practice, ENB can be used for peripheral or central lung nodules. But, CT-guided needle biopsy is only suggested to use for peripheral lung nodule. Therefore, please provide your data about the location of lung nodules.

Response: We agree with the reviewer that location with regards to the periphery is important for TTNA. Since patients were referred to us from different centers, with such a large sample size, there was heterogeneity in radiology reads regarding central vs peripheral location of nodules with a significant number of missing data. Hence we have not included this data and mentioned it in our discussion. 

Reviewer: According to review of literature and my clinical experiences, the sensitivity and specificity of CT-guided needle biopsy were low in ground-glass opacity nodules and small size. Therefore, please clarify the ground-glass opacity of lung nodules in your patients between two groups. I suggested that you should consider the impact of GGO ratio.

Response: That is a great point raised by the reviewer. Since we do not have this data, we have now mentioned that in our limitations.